# Conditional Clifford-Steerable CNNs with Complete Kernel Basis for PDE Modeling

## Abstract

Clifford-Steerable CNNs (CSCNNs) provide a unified framework that allows incorporating equivariance to arbitrary pseudo-Euclidean groups, including isometries of Euclidean space and Minkowski spacetime. In this work, we demonstrate that the kernel basis of CSCNNs is not complete, thus limiting the model expressivity. To address this issue, we propose Conditional Clifford-Steerable Kernels, which augment the kernels with equivariant representations computed from the input feature field. We derive the equivariance constraint for these input-dependent kernels and show how it can be solved efficiently via implicit parameterization. We empirically demonstrate an improved expressivity of the resulting framework on multiple PDE forecasting tasks, including fluid dynamics and relativistic electrodynamics, where our method consistently outperforms baseline methods.

## 1 Introduction

Physical systems exhibit fundamental symmetries that constrain their governing equations (Wang, 2021). Any physically trustworthy model must therefore respect these symmetries and be consistent under the relevant transformations. When these transformations form a symmetry group, models that satisfy this consistency constraint are called *equivariant* (Bronstein et al., 2021). Equivariance has become a key inductive bias in critical applications where adherence to physical laws is required, such as drug and material design (Kovács et al., 2023).

The deep learning community has developed multiple rich theoretical frameworks for constructing equivariant models (Nyholm et al., 2025). For convolutional neural networks (CNNs), the theory of steerable CNNs (Cohen & Welling, 2017) provides a general approach for incorporating equivariance to translations and transformations from an origin-preserving group $G$. A common example is the Euclidean group $E(n)$, where origin-preserving transformations are reflections and rotations that constitute the orthogonal group $O(n)$. The key idea behind steerable CNNs is to leverage the translation-equivariance of convolution and achieve $G$-equivariance by enforcing a $G$-induced constraint on the convolutional kernels. However, solving this constraint is not trivial and is usually involved. To alleviate this complexity, Zhdanov et al. (2023) proposed implicit parameterization using $G$-equivariant MLPs, with the resulting convolutions theoretically guaranteed to be $G$-equivariant.

Zhdanov et al. (2024) further generalized steerable CNNs to pseudo-Euclidean spaces, such as Minkowski spacetime, through Clifford-Steerable CNNs (CSCNNs). CSCNNs process multivector fields by implementing $G$-steerable kernels implicitly via Clifford group equivariant neural networks, leveraging the correspondence between Clifford algebra and pseudo-Euclidean groups. The resulting framework demonstrates strong performance on multiple PDE forecasting tasks; however, it suffers from an inherent limitation: the incompleteness of their steerable kernel basis. Compared to theoretically derived kernel bases, certain degrees of freedom are missing. Although the authors demonstrated that those can be recovered through the use of consecutive convolutions, this solution reveals a fundamental weakness: single CSCNN layers lack expressiveness, thus constraining the model's efficiency and overall performance.

The main contributions of this work are the following:

- We propose *Conditional Clifford Steerable CNNs (C-CSCNNs)*, which remedy the limited expressiveness of original CSCNNs by augmenting $G$-steerable kernels with auxiliary variables derived from input feature fields.

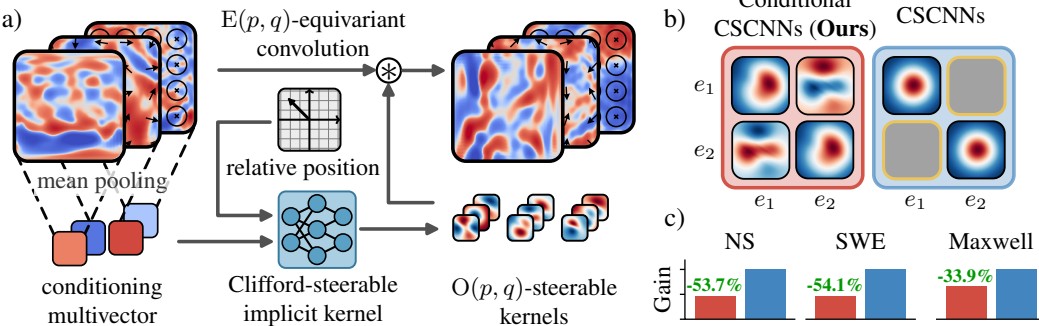

Figure 1: Conditional Clifford-Steerable CNNs use auxiliary information derived from the input feature field to condition the implicit kernel generating $O(p, q)$-steerable kernels (a). The interaction of the additional features with relative positions remedies limited expressivity of the original approach, yielding richer kernel basis (b) and, consequently, substantial performance gains (c).

- We mathematically derive the steerability constraint that conditional kernels must satisfy to maintain $G$-equivariance.
- We demonstrate how this constraint can be solved efficiently via implicit parameterization.
- We empirically validate the improved expressiveness on multiple PDE forecasting tasks, where conditional CSCNNs significantly outperform baseline CSCNNs and achieve performance on par with state-of-the-art methods.

## 2 RELATED WORKS

**Equivariance** There exist multiple ways of incorporating equivariance in neural networks. Equivariant convolutions can be categorized as regular (Cohen & Welling, 2016; Bekkers et al., 2018; 2024) or steerable group convolutions (Weiler & Cesa, 2019; Cesa et al., 2022). Regular group convolutions rely on discretizing the group $G$ and convolving with rotated/transformed versions of the filters. While generally fast, this framework only achieves approximate equivariance due to discretization and is limited to compact groups. Steerable group convolutions employ analytically derived kernels that exactly satisfy the $G$-equivariance constraint. They require knowing the irreducible representation of $G$ along with the Clebsch-Gordan coefficients and harmonic basis functions, all of which are group-specific. Another approach is canonicalization (Mondal et al., 2023; Kaba et al., 2023), which uses a non-equivariant model as the computational backbone but first transforms (canonicalizes) the input to a fixed frame of reference, ensuring that the output transforms consistently. While it demonstrated promising results and integrates well with existing large-scale models, it requires designing a suitable canonicalization procedure, which is in general non-trivial.

**Clifford Algebra** Neural networks based on Clifford algebra were recently popularized by Brandstetter et al. (2023) and have since then gained popularity (Ruhe et al., 2023c; Brehmer et al., 2023) for their ability to natively handle geometric information. Ruhe et al. (2023a) further demonstrated the connection between the Clifford algebra and equivariance, achieving equivariance to the pseudo-Euclidean group $E(p, q)$. Building on this work, Zhdanov et al. (2024) generalized the idea to a convolutional neural network that demonstrated particularly good performance in PDE modelling. Other applications include fluid dynamics (Pepe et al., 2025), relativistic physics (Spinner et al., 2024) and molecular generation (Liu et al., 2025).

**Implicit (continuous) kernels** Parameterizing convolutional kernels as learnable functions that map coordinates to kernel values was initially proposed by Schütt et al. (2017) to handle irregularly sampled molecular data, which later gained popularity in handling general point cloud data (Fuchs et al., 2020; Thomas et al., 2018). A subsequent line of work extended the concept to sequence modelling due to its ability to natively handle irregularly sampled data (Romero et al., 2022; Sitzmann et al., 2020) and capture long-range dependencies (Knigge et al., 2023), thus making them particularly effective in applications to text and audio. Multiple approaches also explored using continuous kernels for equivariant architectures, e.g. by relaxing the equivariance property (van der Ouderaa et al., 2022) or simplifying imposing it (Zhdanov et al., 2023).

## 3 THEORETICAL BACKGROUND

### 3.1 STEERABLE CNNs

We are interested in linear maps that operate on feature (vector) fields over pseudo-Euclidean space $\mathbb{R}^{p,q}$. A feature field is a function $f : \mathbb{R}^{p,q} \rightarrow W$ that assigns a feature vector $f(x) \in W$ to each point $x \in \mathbb{R}^{p,q}$, where $W$ is a vector space. The standard building block in deep learning for operating on such data is a convolutional neural layer, which we define as follows:

**Definition 3.1** (Convolution). Convolution maps between feature fields $f_{\text{in}} : \mathbb{R}^{p,q} \rightarrow W_{\text{in}}$ and $f_{\text{out}} : \mathbb{R}^{p,q} \rightarrow W_{\text{out}}$ via the integral transform

$$f_{\text{out}}(x) = L_K\left[f_{\text{in}}\right](x) := \int_{\mathbb{R}^{p,q}} d\mu(y)\, K(x-y) f_{\text{in}}(y), \tag{1}$$

where $\mu$ is the Lebesgue measure on $\mathbb{R}^{p,q}$, and the kernel $K : \mathbb{R}^{p,q} \rightarrow \text{Hom}(W_{\text{in}}, W_{\text{out}})$ is a function that assigns to each spatial offset $z \in \mathbb{R}^{p,q}$ a linear map $K(z) : W_{\text{in}} \rightarrow W_{\text{out}}$.

We are now interested in enforcing the $G$-equivariance property on the convolutional operator. Before doing so, however, we must describe how feature fields are transformed by elements of $G$. Intuitively, transformations of the base space, which in our case is $\mathbb{R}^{p,q}$, imply corresponding transformations of the feature fields defined on them. To achieve this, we equip feature fields with a $G$-representation $\rho$, which, together with $W$, defines the geometric type of a feature vector $(W, \rho)$ that fully describes the transformation law of a feature field under group action:

$$[\rho(g)f](x) := \rho(g)f\left(g^{-1}x\right) \qquad \forall g \in G, x \in \mathbb{R}^{p,q}. \tag{2}$$

**Definition 3.2** (Equivariance). Consider arbitrary input and output feature fields of types $(W_{\text{in}}, \rho_{\text{in}})$ and $(W_{\text{out}}, \rho_{\text{out}})$. The convolutional operator mapping between them is $G$-equivariant if and only if it satisfies

$$L_K\left[\rho_{\text{in}}(g)f_{\text{in}}\right] = \rho_{\text{out}}(g)L_K\left[f_{\text{in}}\right] \qquad \forall g \in G \tag{3}$$

**Example 3.1.** The map in Def. 3.1 is translation equivariant. We can prove this by taking an arbitrary translation vector $h$ and substituting $\rho_{\text{in}}(h) = \rho_{\text{out}}(h) = T_h$:

$$L_K\left[T_h f_{\text{in}}\right] = \int_{\mathbb{R}^{p,q}} d\mu(y)\, K(x-y) f_{\text{in}}(y-h) = \int_{\mathbb{R}^{p,q}} d\mu(z)\, K((x-h)-z) f_{\text{in}}(z) = T_h L\left[f_{\text{in}}\right]$$

where we used the substitution $y - h \mapsto z$ and the translation invariance property of $\mu$.

The theory of steerable CNNs (Weiler et al., 2023) is concerned with imposing additional equivariance to an arbitrary group $G$ by imposing the following constraint on the kernels:

**Theorem 3.1** (Steerable Convolution, (Weiler et al., 2023)). *The convolution integral in Def. 3.1 is $G$-equivariant if the kernel $K$ satisfies the $G$-steerability constraint*

$$K(gx) = \frac{1}{|\det(g)|}\rho_{\text{out}}(g)K(x)\rho_{\text{in}}(g)^{-1} = \rho_{\text{Hom}}(g)K(x) \qquad \forall g \in G, x \in \mathbb{R}^{p,q} \tag{4}$$

*for feature fields of types $(W_{\text{in}}, \rho_{\text{in}})$ and $(W_{\text{out}}, \rho_{\text{out}})$, and* $|\det(g)|$ *is a volume scaling factor*[1].

In general, one has to solve for the kernel analytically or numerically for each $G$, which is often nontrivial. Moreover, as we will see later, if we were to condition the kernel on an auxiliary variable, we would have to solve the constraint for each such case, which would be practically infeasible. A simple way to circumvent this complexity was suggested by Zhdanov et al. (2023), which boils down to parameterizing a kernel as a continuous function that returns the kernel matrix for each input. The following lemma defines the condition that such a function must satisfy for the resulting kernels to be steerable.

**Lemma 3.1** (Implicit parameterization, (Zhdanov et al., 2023)). *Assume that the linear map $K(x) : W_{\text{in}} \rightarrow W_{\text{out}}$ has matrix representation $[K(x)] \in \mathbb{R}^{c_{\text{out}} \times c_{\text{in}}}$. Let $\phi : \mathbb{R}^{p,q} \rightarrow \mathbb{R}^{c_{\text{out}} \times c_{\text{in}}}$ be a $G$-equivariant function satisfying*

$$\phi(gx) = (\rho_{\text{in}}(g) \otimes \rho_{\text{out}}(g))\phi(x) \qquad \forall g \in G, x \in \mathbb{R}^{p,q}. \tag{5}$$

*Then a kernel parameterized as $[K(x)] = \phi(x)$ satisfies the equivariance constraint in Eq. 4.*

---

[1]For compact groups, e.g. O(2), the factor is 1 and is usually omitted in literature.

## 3.2 Clifford-Steerable CNNs

Implicit parameterization enables solving the $O(p, q)$-steerability constraint allowing implementation of $E(p, q)$-equivariant convolutions (Zhdanov et al., 2024). The work is built on the connection between the pseudo-Euclidean group $E(p, q)$ and the Clifford algebra. Clifford algebra comes with a bilinear operation called the geometric product. By taking two vectors from the vector space $\mathbb{R}^{p,q}$, and taking their product, one obtains multivectors - elements of the $Cl(\mathbb{R}^{p,q})$ algebra. The basis elements of multivectors are $k$-vectors that include scalars ($k = 0$), vectors ($k = 1$), bivectors ($k = 2$), etc. Importantly, Clifford algebra $Cl(\mathbb{R}^{p,q})$ is a representation space of the pseudo-orthogonal group $O(p, q)$ (Zhdanov et al., 2024). Multivectors are associated with the orthogonal representation $\rho_{Cl}$ and thus can be used as features of $O(p, q)$-equivariant networks (Ruhe et al., 2023b).

CSCNNs use such a network for implicit parameterization of the kernels of a $E(p, q)$-equivariant convolution, which operates on $c$-dimensional multivector feature fields $f : \mathbb{R}^{p,q} \to Cl(\mathbb{R}^{p,q})^c$ of type $(W, \rho) = (Cl(\mathbb{R}^{p,q})^c, \rho_{Cl}^c)$. Specifically, the implicit kernel $K$ is a composition of two operators: kernel network $\mathcal{K}$ that returns the multivector-valued matrix representation of the kernel

$$\mathcal{K} : \mathbb{R}^{p,q} \to Cl(\mathbb{R}^{p,q})^{c_{out} \times c_{in}}, \tag{6}$$

and a *kernel head* $H$ that turns the output of $\mathcal{K}$ into a proper steerable kernel

$$H : Cl(\mathbb{R}^{p,q})^{c_{out} \times c_{in}} \to Hom(Cl(\mathbb{R}^{p,q})^{c_{in}}, Cl(\mathbb{R}^{p,q})^{c_{out}}). \tag{7}$$

by partially evaluating the geometric product between input and output feature fields:

$$K = H \circ \mathcal{K}, \qquad K_n^k(x) = \sum_m \Lambda_{mn}^k w_{mn}^k \mathcal{K}(x)^{(m)} \tag{8}$$

where $k$ is the input grade, $n$ is the output grade, $\Lambda_{mn}^k \in \{-1, 0, 1\}$ indicates how the geometric product between the grades $m$ and $n$ manifests in grade $k$ of the result, $w_{mn}^k$ is a learnable weight.

## 3.3 Incompleteness of the kernel basis of CSCNNs

Zhdanov et al. (2024) also showed that, when compared to the analytical solution for $G = O(2) \equiv O(2, 0)$, Clifford-steerable kernels miss degrees of freedom corresponding to angular frequency 2 for the vector-vector field interaction. Therefore, a single layer of CSCNNs is over-constrained by construction, which results in limited expressivity.

The source of the incompleteness lies in the input type of the kernel $K$, and how it is transformed throughout the kernel network $\mathcal{K}$. To demonstrate it, let us consider the case of $O(2, 0)$, for which the analytical solution is derived (Weiler & Cesa, 2019).

**Example 3.2.** The kernel $K$ takes a single vector $x \in \mathbb{R}^{2,0}$ that forms the grade-1 component of the input multivector. Inside $\mathcal{K}$, the only transformations that are applied to the vector are (weighted) geometric products and linear grade-wise transformations. Using polar coordinates of $\mathbb{R}^2 \equiv \mathbb{R}^{2,0}$, i.e. a radius $r \in \mathbb{R}_{\geq 0}$ and angle $\phi \in S$, it can be shown (see Appendix A.4 for details) that none of these operations allows angular information $\phi$ to propagate to any grade other than grade-1:

$$\mathcal{K}(x)^{(0)} = \mathbb{R}_0(r), \quad \mathcal{K}(x)^{(1)} = R_1(r)\kappa_1(\phi), \quad \mathcal{K}(x)^{(2)} = 0 \qquad \forall c_{out}, c_{in}$$

where $R_m, \kappa_m$ denote arbitrary nonlinear functions of $r$ and $\phi$, respectively. We are interested specifically in the vector-vector interaction, as it the part that is missing in the kernel basis. Let $k = n = 1$, then $\Lambda_{01}^1 = \Lambda_{21}^1 = 1$, $\Lambda_{11}^1 = 0$, and the sum in Eq. 8 simplifies to

$$K_1^1(r, \phi) = \mathcal{K}(x)^{(0)} + 0 + 0 = \mathbb{R}_0(r) \tag{9}$$

Consequently, the resulting kernel represents scalar multiplication of the input vector with no angular dependence. In contrast, the complete analytical solution contains both frequency-0 and frequency-2 components, as predicted by the Clebsch-Gordan decomposition of $O(2)$-irrep tensor products (Lang & Weiler, 2021). Importantly, the frequency-2 component cannot be generated by the kernel network $\mathcal{K}$ because its operations on single vector inputs restrict angular information to grade-1.

To re-iterate, the incompleteness, at least in the case of $O(2, 0)$, appears because the vector-vector interaction encoded by the implicit kernel collapses to a scalar value that does not contain angular information when only a single vector serves as input to the geometric product-based neural network. Since we cannot change how the geometric product works, our only option is to change the input to the kernel, which is what we propose in the following section.

# 4    CONDITIONAL CLIFFORD STEERABLE CNNs

Instead of using a purely linear convolutional operator, we can introduce the dependency on the input feature field inside the convolutional kernel. This expands the input to a combination of multivectors, enabling the geometric product-based neural network to encode more expressive kernels and address the limitation of geometric product-based implicit kernels discussed above.

## 4.1    CONDITIONAL STEERABLE CONVOLUTION

Conditioning a convolutional kernel on the input yields the following non-linear operator[2]:

**Definition 4.1** (Conditional Convolution). Conditional convolution maps between feature fields[3] $f : \mathbb{R}^{p,q} \to W_{\text{in}}$ and $f_{\text{out}} : \mathbb{R}^{p,q} \to W_{\text{out}}$ via the integral transform

$$f_{\text{out}}(x) = L_{\hat{K}}\left[f\right](x) := \int_{\mathbb{R}^{p,q}} d\mu(y)\, \hat{K}\left(x - y, f(x), f(y)\right) f(y), \tag{10}$$

where $\mu$ is the Lebesgue measure on $\mathbb{R}^{p,q}$, and the kernel $\hat{K} : \mathbb{R}^{p,q} \times W_{\text{in}} \times W_{\text{in}} \to \text{Hom}(W_{\text{in}}, W_{\text{out}})$ is a function that assigns to every combination of spatial offset $z \in \mathbb{R}^{p,q}$ and points $x, y \in \mathbb{R}^{p,q}$ a linear map $\hat{K}\left(z, f(x), f(y)\right) : W_{\text{in}} \to W_{\text{out}}$.

Similarly to steerable CNNs, we enforce the $G$-equivariance property on the convolutional operator via the $G$-steerability constraint on the conditional kernel $\hat{K}$:

**Lemma 4.1** (Steerable Conditional Convolution). *The convolution integral 4.1 is $G$-equivariant if the kernel $\hat{K}$ satisfies the following $G$-steerability constraint*

$$\hat{K}(g(x-y), \rho_{\text{in}}(g)f(x), \rho_{\text{in}}(g)f(y)) = \rho_{\text{Hom}}(g)\hat{K}(x-y, f(x), f(y)) \qquad \forall g \in G, x \in \mathbb{R}^{p,q} \tag{11}$$

*for input and output feature field types $(W_{\text{in}}, \rho_{\text{in}})$ and $(W_{\text{out}}, \rho_{\text{out}})$, respectively.*

We provide the proof in Appendix A.1.

## 4.2    CLIFFORD-STEERABLE CONDITIONAL CONVOLUTION

As in standard CSCNNs, we use implicit parameterization to achieve equivariance, but we need to adapt the kernel structure to accommodate the auxiliary information. The kernel head $H$ is agnostic to the input of the kernel, furthermore, it is only the kernel network $\mathcal{K}$ that we have to change to be compatible with conditional convolution. Let us define conditional kernel network as

$$\hat{\mathcal{K}} : \mathbb{R}^{p,q} \times \text{Cl}(\mathbb{R}^{p,q})^{c_{\text{in}}} \times \text{Cl}(\mathbb{R}^{p,q})^{c_{\text{in}}} \to \text{Cl}(\mathbb{R}^{p,q})^{c_{\text{out}} \times c_{\text{in}}}, \tag{12}$$

which now takes additional multivector features as input. The function is equivariant by definition, as we use a $\text{O}(p, q)$-equivariant network to parameterize it. The equivariance of resulting Clifford-steerable conditional kernels $\hat{K} := H \circ \hat{\mathcal{K}}$ follows from the following lemma:

**Lemma 4.2** (Equivariance of conditional Clifford-steerable kernels). *Any Clifford-steerable conditional kernel $\hat{K} = H \circ \hat{\mathcal{K}}$ is $\text{O}(p, q)$-equivariant w.r.t $\rho(g) = g$ and $\rho_{\text{Hom}}$ as it satisfies Eq. 11.*

We provide the proof in Appendix A.2.

**Corollary 4.1** (Equivariance of Clifford-steerable conditional convolution). *Let $\hat{K} = H \circ \hat{\mathcal{K}}$ be a Clifford-steerable kernel. The corresponding convolution operator $L_{\hat{K}}$ is then $\text{E}(p, q)$-equivariant:*

$$L_{\hat{K}}\left[\rho_{\text{in}}(g)f\right] = \rho_{\text{out}}(g)L_{\hat{K}}\left[f\right] \qquad \forall g \in \text{E}(p, q). \tag{13}$$

---

[2]For a comprehensive theoretical study of non-linear equivariant operators, see (Nyholm et al., 2025).

[3]In this section, we omit the subscript in $f_{\text{in}}$ for clarity.

## 4.3 Efficient Implementation via Template Matching

The theory admits arbitrary conditioning of the kernel $\hat{K}$ on the input feature field. In the most general case, these might be features at the evaluation $x$ and integration $y$ points. This is similar to message-passing networks or point convolutions, where messages from neighbors to the origin node are typically conditional on the relative distance, as well as features of both sender and receiver nodes. However, such convolution is notoriously slow as it does not have any parameter sharing, the property that makes convolution efficient via template matching. We are furthermore looking for a trade-off between full expressivity (learning arbitrary interactions between two points) and efficiency (fully parallel computation).

To enable template matching, we must condition the kernel on a constant translation-invariant field derived from the input feature field. Let us introduce an operator $T$ that does exactly that

$$T : L^2(\mathbb{R}^{p,q}, \mathrm{Cl}(\mathbb{R}^{p,q})^c) \to \mathrm{Cl}(\mathbb{R}^{p,q})^c, \qquad T[f](x) = T[f] \qquad \forall x \in \mathbb{R}^{p,q}, \qquad (14)$$

and whose output we use as the auxiliary input to Clifford-steerable conditional kernel.

Then the conditional convolution in Def. 4.1 takes the following form:

$$f_{\mathrm{out}}(x) = L^T_{\hat{K}}[f](x) := \int_{\mathbb{R}^{p,q}} d\mu(y)\, \hat{K}(x - y, T[f])\, f(y). \qquad (15)$$

For the convolution to be equivariant, the implicit kernel that parameterizes it must be equivariant as well (Lemma 3.1), which in turn requires equivariance of the operator $T$:

**Proposition 4.1.** *The operator $L^T_{\hat{K}}$ with a Clifford-steerable conditional kernel $\hat{K} = H \circ \hat{\mathcal{K}}$ is $E(p,q)$-equivariant if the operator $T$ is $O(p,q)$-equivariant, i.e.*

$$T[\rho(g)f] = \rho(g)T[f] \qquad \forall g \in O(p,q), f \in L^2(\mathbb{R}^{p,q}, \mathrm{Cl}(\mathbb{R}^{p,q})^c) \qquad (16)$$

We provide the proof in Appendix A.3.

The simplest choice for $T$ is mean pooling whose equivariance is proven in (Weiler et al., 2023). Using mean pooling essentially replaces message-passing form of Eq. 4.1 with a mean-field approximation where each point is influenced by the average behavior of all other points. While less expressive, the formulation nonetheless allows the kernels to adjust to global context, making it strictly more expressive than a standard convolution.

## 4.4 Completeness of the kernel basis of Conditional CSCNNs

Let us now return to the case of O(2,0) and see how it will be handled in the suggested framework.

**Example 4.1** (Continued). Let us assume that the auxiliary input is a single multivector $\zeta \in \mathrm{Cl}(\mathbb{R}^{p,q})$, whose grade-1 component does not coincide with the relative position vector $x \in \mathbb{R}^{p,q}$. It can be shown (see Appendix A.5), that after a single geometric product between $\zeta$ and the embedded $x$, the kernel network output takes the following form

$$\mathcal{K}(x)^{(m)} = \mathbb{R}_m(r)\kappa_1(\phi) \qquad \forall m, c_{\mathrm{out}}, c_{\mathrm{in}}$$

including the grade-0 case, which in the standard case is only a function of $r$. In other words, the angular information propagates through interaction with auxiliary multivectors. This, in turn, allows conditional implicit kernels to replicate the analytical solution as they are now able to generate higher frequency components.

Therefore, it is evident that using conditional kernels alleviates the incompleteness issue as the frequency-2 components are present in their basis. We formulate the result as a conjecture:

**Conjecture 4.1.** *The kernel basis of conditional Clifford-steerable CNNs is complete.*

We note that proving the conjecture requires knowing all irreducible representations of $O(p,q)$ for each combination of $p$ and $q$, as well as constructing an isomorphism between each irrep and multivector grades of $\mathrm{Cl}(p,q)$. The task is highly non-trivial and somewhat contradictory to the nature of implicit kernels, which are designed to avoid deriving steerable kernels analytically. We therefore leave the proof for future work, and validate the claim empirically in the following section.

## 5 EXPERIMENTS

To experimentally validate the improved expressivity of our implementation, we tested its performance on four well-established PDE modeling benchmarks, comparing it to the original CSCNN model and 14 strong baselines. The experiments conducted were the 2-dimensional ($\mathbb{R}^2$) *Navier-Stokes (NS)* equations, the 2-dimensional ($\mathbb{R}^2$) *Shallow-water (SWE)* equations, the 3-dimensional ($\mathbb{R}^3$) *Maxwell's (MW3)* equations, and the relativistic 2-dimensional ($\mathbb{R}^{1,2}$) *Maxwell's (MW2)* equations. Further details on the datasets can be found in Appendix C.

### 5.1 EXPERIMENTAL SETUP

The goal of each experiment is to learn the dynamics of a system from numerical simulations. To enable comparisons, we treat timesteps as feature channels - i.e., the task becomes predicting future states (as output channels) based on previous states (as input channels) (Gupta & Brandstetter, 2022). For NS and SWE, the state is described by a vector velocity and a scalar pressure field. We evaluate two setups of SWE: one-step predictions (SWE-1), and 5-step rollouts (SWE-5). For MW3, the inputs are vector electric fields and bivector magnetic fields. In the relativistic MW2 task, the input is an electromagnetic field, which forms a bivector. Let $N_t$ denote the number of points used to discretise time in each state. In this setting, time forms its own spacetime dimension, therefore, the task is to learn the mapping between two states, each with their respective spatial dimensions and a time dimension of size $N_t$. This is the only experiment where the setup properly incorporates the time dimension into spacetime. Conditional CSCNNs (C-CSCNNs), similarly to regular CSCNNs, are well-equipped to fit this setting, which other baselines are unable to handle by construction.

### 5.2 IMPLEMENTATION

Note that since C-CSCNNs process multivector fields, we embed the feature fields into their corresponding multivector basis elements, which is enabled by the natural isomorphisms $\varepsilon^{(0)} : \mathbb{R} \xrightarrow{\sim} \mathrm{Cl}(\mathbb{R}^{p,q})^{(0)}$, $\varepsilon^{(1)} : \mathbb{R}^{p,q} \xrightarrow{\sim} \mathrm{Cl}(\mathbb{R}^{p,q})^{(1)}$ between scalars, vectors and their corresponding $k = 0$ and $k = 1$ multivector grades. Our model is built upon the ResNet (He et al., 2016) architecture, where we substitute the standard convolutional layers with our Conditional Clifford-Steerable convolutions, where we use global mean pooling as the conditioning operator $T$. Compared to the original CSCNNs, the only additional computational cost derives from computing the condition, which in the case of pooling is negligible. The implementation is done in JAX (Bradbury et al., 2018) and Flax (Heek et al., 2024), leveraging the benefits of XLA compilation.

We compare Conditional CSCNNs against multiple families of strong neural solvers. In the NS, MW3, and MW2 experiments, we evaluate against architec-

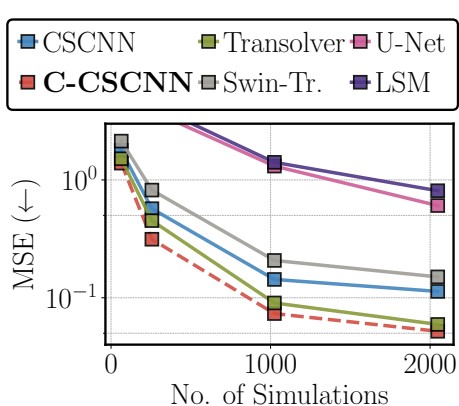

Figure 2: MSE for the Shallow-water equations $\mathbb{R}^2$ 1-step forecasting task as a function of the simulations included in the training dataset. Conditional CSCNNs outperform all baselines, keeping their advantage even as the training trajectories increase.

turally similar networks: the standard ResNet, the Clifford ResNet (Brandstetter et al., 2023), the O(n)-steerable CNN (Weiler & Cesa, 2019; Weiler et al., 2023) and the original Clifford-Steerable CNN (Zhdanov et al., 2024). Additionally, we include Fourier Neural Operator (FNO) (Li et al., 2021) and its $D_4 < O(2)$ equivariant counterpart G-FNO (Helwig et al., 2023) in the evaluation. On SWE-1, the models compared are the state-of-the-art neural PDE solvers: Transolver (Wu et al., 2024a), the Swin-Transformer (Liu et al., 2021), the classic U-net (Ronneberger et al., 2015), the latent-space solver LSM (Wu et al., 2023) and the original Clifford-Steerable CNN (Zhdanov et al., 2024). For SWE-5, we take baselines from (Gupta & Brandstetter, 2022) and (Wang et al., 2025), which also include different sizes of CViT (Wang et al., 2025), the Dilated ResNet (Yu et al., 2017), with two of its variants U-Net$_{att}$ (Gupta & Brandstetter, 2022) and U-F2Net (Gupta & Brandstetter, 2022), FNO (Li et al., 2021) and the U-shaped neural operator UNO (Rahman et al., 2023). In each experiment except SWE-5, the parameter counts were matched to ensure fair comparison

between the models. In SWE-5 however, we compared architectures of different sizes. More details on the tasks and the implementations can be found in Appendix C and B.

## 5.3 RESULTS

**Data efficiency** In the NS, SWE-1, MW3 and MW2 tasks, we investigate the data efficiency of the models by varying the number of trajectories they are trained on. Conditional CSCNNs show exceptional ability in learning system's dynamics, achieving the best results on all four tasks. As shown in Figure 3, the advantage granted by the expressive conditional kernels becomes apparent with only a few training trajectories. Moreover, owing to their improved expressivity, C-CSCNNs are able to leverage more training data significantly better than standard CSCNNs, or other baselines. In the SWE-1 task shown in Figure 2, C-CSCNNs outperform state-of-the-art models such as Transolver or Swin-Transformer. By transforming consistently under the symmetries of the physical system, they are able to effectively capture the true governing dynamics, providing accuracy improvements even in highly transient regions of the simulation domain, visualised in Figure 6.

Table 1: 5-step rollout performance on the shallow-water equations task by Relative $L^2$ error (lower is better.) The results for baselines were taken from (Wang et al., 2025)

| Model | #Params | Rel. $L^2$ |
|---|---|---|
| DilResNet | 4.2M | 13.20% |
| U-Net$_{att}$ | 148M | 5.68% |
| FNO | 268M | 3.97% |
| U-F2Net | 344M | 1.89% |
| UNO | 440M | 3.79% |
| CViT-S | 13M | 4.47% |
| CViT-B | 30M | 2.69% |
| CViT-L | 92M | 1.56% |
| C-CSCNN (Small) | 10M | 3.51% |
| C-CSCNN (Large) | 55M | 2.94% |

**Scaling** For the SWE-5 benchmark, we probed the scaling properties of our approach. Table 1 shows the relative $L^2$ error for the 5 step predictions (see Fig.4 for a rollout example). In the small model regime (around 10M), conditional CSCNNs significantly outperform even state-of-the-art models, such as CViT. When scaled up, despite being built on a simple ResNet architecture, they perform on par with leading approaches and exceed models significantly larger (such as FNO with 270M and UNO with 440M params), showing their potential for larger-scale modeling tasks.

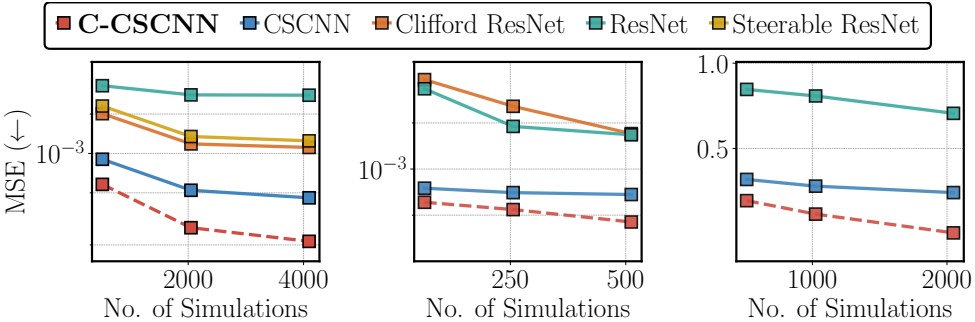

Figure 3: Mean squared errors for (1) Navier-Stokes $\mathbb{R}^2$, (2) Maxwell $\mathbb{R}^3$, and (3) relativistic Maxwell $\mathbb{R}^{1,2}$ simulation tasks as a function of the simulations included in the training dataset. Conditional CSCNNs outperform all baselines, with their advantage increasing as more data is included in the training set.

**Equivariance** To support our theoretical claims in Proposition 4.1, we evaluated the equivariance of our novel convolutional layers against the original Clifford-Steerable convolutions. Table 2 shows the relative $E(2)$ equivariance error

$$\text{err}(f; g, x) = \frac{|f(g \cdot x) - g \cdot f(x)|}{|f(g \cdot x) + g \cdot f(x)|}$$

Table 2: Relative equivariance errors for Clifford-Steerable convolutions.

| Convolution | Relative error (mean) |
|---|---|
| CS Convolution | $2.4 \times 10^{-7}$ |
| **C-CS Convolution** | $3.4 \times 10^{-7}$ |

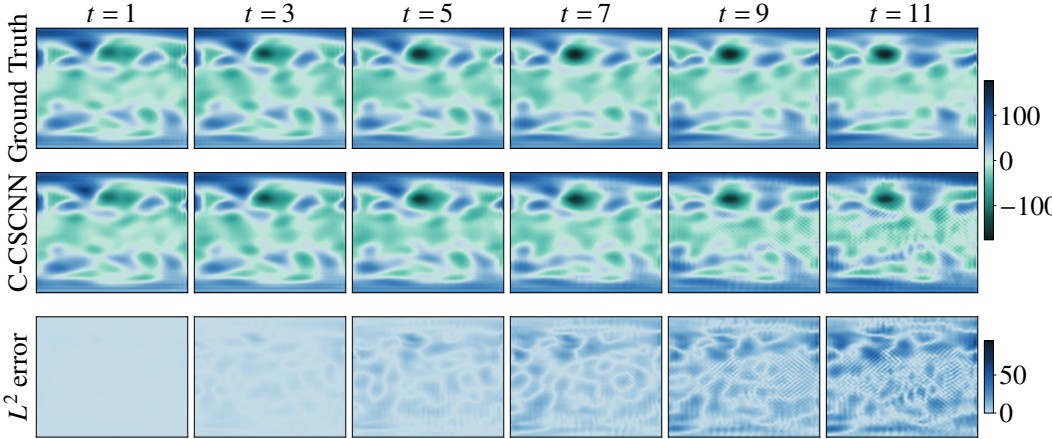

Figure 4: Relative $L^2$ error of conditioned CSCNNs on the shallow water equations task at different steps of the rollout trajectories. Results are shown for component $u$ of the wind velocity field.

for the default, and conditional Clifford-Steerable convolutions. Up to numerical artifacts, conditional convolutions show relative errors similar to the default convolutions, providing an experimental validation of our equivariance proof.

## 6 CONCLUSION

We introduced Conditioned Clifford-Steerable CNNs, a generalization of the Clifford-Steerable CNNs that addresses the limited expressivity of the original framework. Our approach augments the parameterization of Clifford-steerable kernels with auxiliary variables derived from the input, effectively making the convolution operation nonlinear. We derived the equivariance constraint on such kernels and proved that our approach satisfies it. Through comprehensive empirical evaluation on various PDE forecasting tasks, we demonstrated that our approach consistently outperforms the original CSCNNs while achieving strong overall performance compared to state-of-the-art methods.

**Future work** The most promising avenue for future work is exploring different choices of the conditional operator. In this work, we already observed that by using simple mean pooling, we are able to achieve performance comparable to state-of-the-art approaches. Potential options include different types of pooling, e.g. max pooling or learnable pooling. The overall strong performance of our framework suggests that conditioning kernels in convolutions is a sound approach. Although our current implementation relies on mean-field approximations for kernel conditioning, future work can explore less restrictive forms of weight sharing, e.g. conditioning on finer yet still coarse regions of the domain. This would render the approach closer to hierarchical methods such as fast multipole method (Carrier et al., 1988) which recently found application in deep learning (Wang, 2023; Wu et al., 2024b; Zhdanov et al., 2025).

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

# A  APPENDIX: PROOFS

## A.1  PROOF OF LEMMA 4.1

*Proof.* A conditional convolution given in Def. 4.1 is $G$-equivariant if it satisfies

$$L_{\hat{K}}\left[\rho_{\text{in}}(g)f\right] = \rho_{\text{out}}(g)L_{\hat{K}}\left[f\right] \qquad \forall g \in G. \tag{17}$$

Let us expand the left-hand side:

$$L_{\hat{K}}\left[\rho_{\text{in}}(g)f\right] = \int_{\mathbb{R}^{p,q}} d\mu(y)\, \hat{K}\left(x - y, \rho_{\text{in}}(g)f(g^{-1}x), \rho_{\text{in}}(g)f(g^{-1}y)\right)\rho_{\text{in}}(g)f(g^{-1}y) \tag{18}$$

$$= |\det(g)| \int_{\mathbb{R}^{p,q}} d\mu(y)\, \hat{K}\left(x - gy, \rho_{\text{in}}(g)f(g^{-1}x), \rho_{\text{in}}(g)f(y)\right)\rho_{\text{in}}(g)f(y) \tag{19}$$

where we used the substitution $g^{-1}y \mapsto y$

Writing out the right-hand side yields

$$\rho_{\text{out}}(g)L_{\hat{K}}\left[f\right] = \int_{\mathbb{R}^{p,q}} d\mu(y)\, \rho_{\text{out}}(g)\hat{K}\left(g^{-1}x - y, f(g^{-1}x), f(y)\right)f(y) \tag{20}$$

The expressions agree for any $g \in G$ and and feature map $f \in L^2(\mathbb{R}^{p,q}, W)$ if and only if

$$|\det(g)|\hat{K}\left(x - gy, \rho_{\text{in}}(g)f(g^{-1}x), \rho_{\text{in}}(g)f(y)\right)\rho_{\text{in}}(g) = \tag{21}$$

$$\rho_{\text{out}}(g)\hat{K}\left(g^{-1}x - y, f(g^{-1}x), f(y)\right). \tag{22}$$

After the substitution $g^{-1}x \mapsto x$, we obtain the constraint:

$$|\det(g)|\hat{K}\left(gx - gy, \rho_{\text{in}}(g)f(x), \rho_{\text{in}}(g)f(y)\right)\rho_{\text{in}}(g) = \rho_{\text{out}}(g)\hat{K}\left(x - y, f(x), f(y)\right), \tag{23}$$

which is equivalent to

$$\hat{K}(g(x - y), \rho_i(g)f(x), \rho_i(g)f(y)) = \frac{1}{|\det(g)|}\rho_o(g)\hat{K}(x - y, f(x), f(y))\rho_i(g)^{-1} \tag{24}$$

$$= \rho_{\text{Hom}}(g)\hat{K}(x - y, f(x), f(y)). \tag{25}$$

$\square$

## A.2  PROOF OF LEMMA 4.2

*Proof.* A Clifford-steerable conditional kernel is a composition of kernel network $\hat{\mathcal{K}}$ and kernel head $H$. The former is equivariant by definition, and the equivariance of the latter is proven in Zhdanov et al. (2024). We can then write

$$\hat{K}\left(g(x - y), \rho_{\text{Cl}}^{c_{\text{in}}}(g)f(x)\rho_{\text{Cl}}^{c_{\text{in}}}(g)f(y)\right) = H\left(\hat{\mathcal{K}}(g(x - y), \rho_{\text{Cl}}^{c_{\text{in}}}(g)f(x)\rho_{\text{Cl}}^{c_{\text{in}}}(g)f(y)\right) \tag{26}$$

$$= H\left(\rho_{\text{Cl}}^{c_{\text{out}} \times c_{\text{in}}}(g)\hat{\mathcal{K}}(x - y, f(x), f(y)\right) \tag{27}$$

$$= \rho_{\text{Hom}}(g)H\left(\hat{\mathcal{K}}(x - y, f(x), f(y)\right) \tag{28}$$

$$= \rho_{\text{Hom}}(g)\hat{K}\left(x - y, f(x), f(y)\right) \tag{29}$$

$\square$

## A.3  PROOF OF PROPOSITION 4.1

*Proof.* By the definition of the kernel network $\hat{\mathcal{K}}$, it satisfies the equivariance constraint

$$\hat{\mathcal{K}}(gz, \rho(g)w) = (\rho(g) \otimes \rho(g))\hat{\mathcal{K}}(z, w) \qquad \forall g \in \mathrm{O}(p, q), z \in \mathbb{R}^{p,q}, w \in \mathrm{Cl}(\mathbb{R}^{p,q})^c. \tag{30}$$

Furthermore, the property is satisfied for $w = T[f]$:

$$\hat{\mathcal{K}}(gz, \rho(g)T[f]) = (\rho(g) \otimes \rho(g))\hat{\mathcal{K}}(z, T[f]) \tag{31}$$

On the other hand, for the kernel to be steerable, we require:

$$\hat{\mathcal{K}}(gz, T[\rho(g)f]) = (\rho(g) \otimes \rho(g))\hat{\mathcal{K}}(z, T[f]). \tag{32}$$

Comparing these two equations, we conclude that the left hand sides must coincide

$$\hat{\mathcal{K}}(gz, \rho(g)T[f]) = \hat{\mathcal{K}}(gz, \rho(g)T[f]) \tag{33}$$

for all $g \in \mathrm{O}(p,q), x,y \in \mathbb{R}^{p,q}$. There expressions agree if and only if

$$\rho(g)T[f] = T[\rho(g)f] \tag{34}$$

which is the equivariance constraint on the operator $T$. $\qquad\square$

## A.4 INCOMPLETENESS OF CSCNNs; $\mathrm{O}(2,0)$

Since the grade mixing in CEGNNs happens exclusively via geometric product, it is sufficient to demonstrate how multivector encoding relative position multiplies with itself. We use sympy (Meurer et al. (2017)) to do the algebraic manipulations. The code snippet is provided in Code 1.

```python
import sympy

# relative position in polar coordinates
r, phi = sympy.symbols('r phi', real=True)
# embedding as a Clifford multivector
r_clifford = [r, r*sympy.cos(phi), r*sympy.sin(phi), 0]
# compute geometric product and print the output
k = geometric_product(r_clifford, r_clifford)

print(f"{k[0]} + {k[1]}e1 + {k[2]}e2 + {k[3]}e1e2")
# 2*r**2 + 2*r**2*cos(phi)e1 + 2*r**2*sin(phi)e2 + 0e1e2
```

Code 1: Geometric product of a multivector with itself does not propagate angular information $\phi$.

## A.5 COMPLETENESS OF CONDITIONAL CSCNNs; $\mathrm{O}(2,0)$

Similarly, it can be demonstrated that for the case of two different multivectors, one of which encodes relative position, the output of the geometric product does contain angular information $\phi$ in the scalar part, which is the necessary part to replicate the analytical solution of $\mathrm{O}(2)$-steerable kernels. The code snippet is provided in Code 2. By increasing the number of layers in CEGNN, and the number of auxiliary multivector, we are able to recover high-frequency functions in the scalar part (see Code 3).

```python
import sympy

# relative position in polar coordinates
r, phi = sympy.symbols('r phi', real=True)
# elements of the second multivector
x0, x1, x2, x3 = sympy.symbols('x0 x1 x2 x3', real=True)
# embedding as a Clifford multivector
r_clifford = [r, r*sympy.cos(phi), r*sympy.sin(phi), 0]
# second multivector
x_clifford = [x0, x1, x2, x3]
# compute geometric product and print the output
kx = geometric_product(r_clifford, x_clifford)

print(f"{kx[0]} + {kx[1]}e1 + {kx[2]}e2 + {kx[3]}e1e2")
# r*(x0 + x1*cos(phi) + x2*sin(phi)) +\
# r*(x0*cos(phi) + x1 - x3*sin(phi))e1 +\
# r*(x0*sin(phi) + x2 + x3*cos(phi))e2 +\
# r*(-x1*sin(phi) + x2*cos(phi) + x3)e1e2
```

Code 2: Geometric product of two multivectors allows angular information to propagate.

```
1  # elements of the third multivector
2  y0, y1, y2, y3 = sympy.symbols('y0 y1 y2 y3', real=True)
3  ky = geometric_product(r_clifford, y_clifford)
4
5  # compute geometric product between the outputs (2nd layer)
6  k = geometric_product(kx, ky)
7
8  # print the scalar part of the output
9  print(kx[0])
10 # r**2*( 2*x0*y0 + + x2*y2 +\
11 #        2*x0*y1*cos(phi) + 2*x0*y2*sin(phi) +\
12 #        2*x1*y0*cos(phi) + 2*x2*y0*sin(phi) + \
13 #        x1*y1*cos(2*phi) + x1*y1 + x1*y2*sin(2*phi) +\
14 #        x2*y1*sin(2*phi) - x2*y2*cos(2*phi)
15 #)
```

Code 3: Increasing the number of layers and auxiliary multivectors allows computing high-frequency components.

## B    APPENDIX: IMPLEMENTATION DETAILS

The basic ResNet architecture that was used for constructing the Conditioned-CSCNN, the CSCNN, the Clifford ResNet and the O($n$)-steerable ResNet were based on the setup of Wang et al. (2021); Brandstetter et al. (2023); Gupta & Brandstetter (2022). They consist of 8 residual blocks with $7 \times 7$ and $7 \times 7 \times 7$ sized kernels for the 2D and 3D experiments respectively. We used two embedding and two output layers. When constructing the other baseline models, we closely followed the original implementations discussed in their respective papers, which are referenced in 5.2. The models have approximately 7M parameters for the Navier Stokes, 7.5M for the one step Shallow-waters and 1.5M for Maxwell's experiments. To scale up C-CSCNN to 55M parameters in the SWE-5 task, we increased the number of residual blocks to 12, and the number of hidden channels from 48 to 96. Similarly, we increased the hidden dimensions of the parametrising kernel network from 12 to 16, and increased its depth from 4 to 6 layers.

### B.0.1    CONDITIONAL KERNEL IMPLEMENTATION

In constructing the Conditioned Clifford-Steerable Kernels, we built on the architecture described in Appendix A of Zhdanov et al. (2024). We form the conditioning vector by a *masked spatial mean* computed for each channel $c$ and blade/grade $k$. On a grid $\Omega \subset \mathbb{R}^d$ and with the indicator $\chi_{B_r}$ of the largest centered ball $B_r$ (the *circular / spherical* mask, which we explain below), the pooled stack is

$$\left(T_{\text{pool}}[f_{\text{in}}]\right)_c^{(k)} := \frac{1}{|\Omega|} \sum_{x \in \Omega} \chi_{B_r}(x) \, f_{\text{in},c}^{(k)}(x),$$

The resulting conditioning multivector stack is then concatenated with the relative position vector to form the input to the Kernel Network.

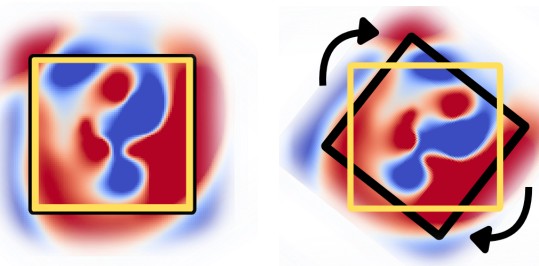

Figure 5: Illustration of how the receptive field of a finite, discretized kernel changes under rotations. The square support causes operations to break equivariance near the corners.

**Circular mask** :    In practice, the multivector feature fields are often discretised as square/rectangular-shaped arrays $(c, X_1 \ldots X_d, 2^d)$. Thus, an operation defined solely on this finite grid will break equivariance towards the corners of the domain, as shown in Figure 5. To overcome this, we apply circular masking (set grid values outside of the circle/sphere to 0) before the pooling operation, making it O($n$)-equivariant in the continuum.

**Normalisation**   To avoid instability in the early phases of training, we employ learnable, equivariant grade-wise normalisation of the conditioning vector stack. Note that this was only necessary for the MW2 task.

### B.0.2   TRAINING DETAILS

For the training, we adopted the optimised hyperparameters from Zhdanov et al. (2024) for our models in the NS, MW3 and MW2 experiments. For SW2-1, we adopted hyperparameters from the Transolver paper (Wu et al., 2024a). We used Adam optimizer (Kingma & Ba, 2015) with cosine learning rate scheduler for SWE with, and for MW3 and MW2 without warmup. Each model was trained to convergence. The models were trained on one node of Snellius, the Dutch national computing cluster with 1-4 NVIDIA A100 GPUs.

## C   APPENDIX: DATASETS

**2D Navier Stokes equations**   The ground truth simulations are taken from Gupta & Brandstetter (2022) and are based on $\Phi$Flow by Holl & Thuerey (2024). From the corresponding validation and test partitions, we randomly sampled $1024$ trajectories. The simulations were generated on a $128 \times 128$ pixel grid with uniform spatial spacing $\Delta x = \Delta y = 0.25\,\mathrm{m}$ and a time step of $\Delta t = 1.5\,\mathrm{s}$. The task consists of predicting the proceeding timestep based on the previous 4 states.

**2D Shallow-water equations**   The dataset os obtained from Gupta & Brandstetter (2022), who utilisied an implementation of `SpeedyWeather.jl` (Klöwer et al., 2024). The simulations are performed on a grid with spatial resolution of $192 \times 96$, $\Delta x = 1.875°$, $\Delta y = 3.75°$ and temporal resolution of $\Delta t = 48$h. The equations evaluated are in velocity function formulation, with vector wind velocity and scalar pressure fields to be predicted. We sampled trajectories from the validationa and test partitions randomly. For the SWE-1 task, the domain was cropped and downsampled to a grid of $64 \times 64$ spatial points to speed up training. The task for SWE-1 is to predict the next timestep given the previous 4, whereas for SWE-5, the proceeding 5 states are to be predicted, given the previous 2. The final dataset for SWE-1 consisted of 512 test, 512 valid, and max 2048 train trajectories. For SWE-5, the entire train, test, valid dataset, described in Gupta & Brandstetter (2022) was used.

**3D Maxwell's equations**   Within the non-relativistic $\mathrm{Cl}(\mathbb{R}^{3,0})$ setting, we represent the electric field $E$ as a vector field and the magnetic field $B$ as a bivector field. The dataset, drawn from Brandstetter et al. (2023), consists of 3D Maxwell simulations discretized on a $32 \times 32 \times 32$ voxel grid with uniform spacing $\Delta x = \Delta y = \Delta z = 5 \times 10^{-7}\,\mathrm{m}$ and time step $\Delta t = 50\,\mathrm{s}$. The validation and test splits each contain 128 simulations. Similarly to NS2, the task consists of predicting the proceeding timestep based on the previous 4 states.

**2D relativistic Maxwell's equations**   We generate a dataset for Maxwell's equations in 2+1D spacetime ($\mathbb{R}^{1,2}$) utilizing the `PyCharge` simulation package by Filipovich & Hughes (2022). The simulations model the dynamics of electromagnetic fields emitted by oscillating and orbiting point charges moving at relativistic speeds. The spacetime grid is discretized with a resolution of $128 \times 128$ points, corresponding to a spatial extent of $50\,\mathrm{nm}$ and a temporal duration of $3.77 \cdot 10^{-14}\,\mathrm{s}$.

Each simulation is initialized with a unique configuration of charge sources, governed by the following randomly sampled parameters: **Source Composition:** A combination of 2 to 4 oscillating charges and 1 to 2 orbiting charges, with integer magnitudes sampled uniformly from the range $[-3e, 3e]$. **Initial Conditions:** Sources are placed uniformly on the grid with a predefined minimum separation. Each is assigned a random linear velocity and either oscillates in a random direction or orbits with a random radius. **Relativistic Constraint:** Oscillation/rotation frequencies and velocities are sampled such that the total particle velocity does not exceed $0.85c$, a necessary constraint to ensure the stability of the PyCharge solver.

To handle the wide dynamic range of the resulting field strengths, we apply a normalization scheme. The generated field bivectors are divided by their Minkowski norm and then multiplied by the logarithm of that norm. Although Minkowski norms can be zero or negative, we found they were consistently positive in our generated data. Finally, we filter numerical artifacts by removing any

outlier simulations that exhibit a standard deviation greater than 20. The curated dataset is split into 2048 training, 256 validation, and 256 test simulations.

## D  APPENDIX: EXPERIMENTAL RESULTS

We provide a comprehensive overview of the experiment results in Table 3.

### D.0.1  ERROR COMPARISON



Figure 6: Signed residuals (prediction−ground truth) of wind velocity vector component $u$ in the one step ahead predictions for Shallow-Water Equations $\mathbb{R}^2$.

| NS | $N = 64$ | $N = 512$ | $N = 2048$ | $N = 4096$ |
|---|---|---|---|---|
| **C-CSCNN** | **2.38e-03** | **5.80e-04** | **2.70e-04** | **1.80e-04** |
| Transolver | 2.49e-03 | 7.10e-04 | 4.10e-04 | 3.40e-04 |
| CSCNN | 2.63e-03 | 9.00e-04 | 5.30e-04 | 4.50e-04 |
| Clifford ResNet | 4.21e-03 | 2.01e-03 | 1.18e-03 | 1.11e-03 |
| Steerable ResNet | 8.41e-03 | 2.30e-03 | 1.35e-03 | 1.25e-03 |
| ResNet | 4.57e-03 | 3.29e-03 | 2.80e-03 | 2.79e-03 |
| CViT | 0.0121 | 2.15e-03 | 1.01e-03 | 9.45e-04 |
| G-FNO | 0.0187 | 6.77e-03 | 5.41e-03 | 4.90e-03 |
| FNO | 0.0317 | 0.0116 | 8.70e-03 | 8.20e-03 |
| **SWE-1** | $N = 64$ | $N = 256$ | $N = 1024$ | $N = 2048$ |
| **C-CSCNN** | **1.3886** | **0.3137** | **0.0734** | **0.0519** |
| Transolver | 1.5127 | 0.4515 | 0.0903 | 0.059404 |
| CSCNN | 1.7829 | 0.5699 | 0.1431 | 0.1132 |
| CViT | 1.9871 | 0.7129 | 0.1974 | 0.12589 |
| Swin-Tr. | 2.1341 | 0.8197 | 0.2073 | 0.1503 |
| U-Net | 8.5385 | 3.8517 | 1.3156 | 0.6057 |
| LSM | 9.1869 | 4.3282 | 1.4192 | 0.8094 |
| **MW3** | $N = 64$ | $N = 256$ | $N = 512$ | |
| **C-CSCNN** | **6.07e-04** | **5.44e-04** | **4.51e-04** | |
| CSCNN | 7.50e-04 | 7.02e-04 | 6.82e-04 | |
| ResNet | 3.34e-03 | 1.90e-03 | 1.67e-03 | |
| Clifford ResNet | 3.83e-03 | 2.57e-03 | 1.70e-03 | |
| FNO | 0.0191 | 0.0166 | 0.0141 | |
| **MW2** | $N = 512$ | $N = 1024$ | $N = 2048$ | |
| **C-CSCNN** | **0.3273** | **0.2934** | **0.2519** | |
| CSCNN | 0.3889 | 0.3682 | 0.3494 | |
| ResNet | 0.8081 | 0.7661 | 0.6653 | |
| **SWE-5** | *Rel. $L^2$ Error (5-step rollout)* | | | |
| *Model (Params)* | | | | |
| CViT-L (92M) | **1.56%** | | | |
| U-F2Net (344M) | 1.89% | | | |
| CViT-B (30M) | 2.69% | | | |
| C-CSCNN (Large) (55M) | 2.94% | | | |
| C-CSCNN (Small) (10M) | 3.51% | | | |
| UNO (440M) | 3.79% | | | |
| FNO (268M) | 3.97% | | | |
| CViT-S (13M) | 4.47% | | | |
| U-Net$_{att}$ (148M) | 5.68% | | | |
| DilResNet (4.2M) | 13.20% | | | |

Table 3: Comparison of MSE across different experiments and training dataset sizes (the latter denoted by $N$ in the experiment titles). We report relative L2 error for the SWE-5 task to enable easy to results reported in corresponding papers. Best results per column are bolded.

