# OpenReview forum: "Conditional Clifford-Steerable CNNs with Complete Kernel Basis for PDE Modeling"
_ICLR.cc/2026/Conference — Submitted to ICLR 2026_

### Official Review · Reviewer_3dR3 · 2025-10-30

**Soundness:** 2
**Presentation:** 3
**Contribution:** 1
**Rating:** 2
**Confidence:** 5

**Summary:**

This paper proposes Conditional Clifford-Steerable CNNs (C-CSCNNs), an extension of Clifford-Steerable CNNs (CSCNNs). The core idea is to address the incomplete kernel basis of the original CSCNN framework by augmenting the steerable kernels with auxiliary variables derived from the input feature field. The authors claim this "conditioning" mechanism completes the kernel basis, thereby enhancing the model's expressivity. They derive the equivariance constraint for these input-dependent kernels and demonstrate their method's effectiveness on several PDE forecasting tasks, where it reportedly outperforms baseline methods.

**Strengths:**

*   **Significance of the Problem:** The paper correctly identifies a significant limitation in the CSCNN framework (Zhdanov et al., 2024)—the incompleteness of its kernel basis. Addressing this limitation is a worthwhile endeavor, as a complete basis would theoretically lead to a more expressive and powerful equivariant model, which is crucial for accurately modeling physical systems with complex symmetries.
*   **Clarity of Motivation:** The introduction clearly articulates the problem. The authors provide a good intuition in Section 3.3 and Example 3.2 on why the original CSCNNs fail to generate certain kernel components (e.g., frequency-2 components for vector-vector interactions in $O(2,0)$), which effectively motivates the need for a solution.

**Weaknesses:**

This paper, while addressing an important problem, suffers from significant weaknesses in its algorithmic contribution and, most critically, in its experimental validation. These issues make it impossible to fairly assess the true effectiveness of the proposed method.

1.  **Limited Algorithmic Contribution:** The paper is heavily laden with mathematical formalism, much of which is directly inherited from the prior work on CSCNNs and general steerable CNN theory. The actual novel contribution is the "conditioning" mechanism, which is implemented in a very straightforward manner: using global mean pooling to generate a single conditioning multivector. While the derivation of the corresponding equivariance constraint (Lemma 4.1) is technically sound, the core algorithmic idea itself is an incremental extension rather than a substantial leap forward. The heavy reliance on existing mathematical frameworks makes the paper's own contribution appear smaller than presented.

2.  **Insufficient and Inconsistent Experimental Setup:** This is the most critical flaw of the paper. The experimental setup is confusing, inconsistent, and incomplete, which severely hinders the reader's ability to verify the authors' claims.
    *   **Inconsistent Baselines and Presentation:** The paper claims to compare against 14 baselines, but this is not reflected consistently in the results. Figure 2 shows only 5 baselines. Table 1 includes 8 baselines, with the data admittedly being taken from another paper (Wang et al., 2025), which is not a direct, controlled comparison. Figure 3 shows only 6 baselines. Furthermore, the set of baselines changes from one experiment to another. This inconsistency makes it impossible to draw a clear, overarching conclusion about the method's performance relative to the state-of-the-art.
    *   **Lack of Standardized Benchmarking:** To ensure a fair and reproducible comparison, the authors should evaluate their model on a standardized benchmark like PDEBench. I strongly recommend that the authors re-run their experiments and present the main results in a comprehensive table. This table should include, for all datasets, a consistent set of strong baselines, such as **Transolver, the original CSCNN, Swin-Transformer, CViT-L, and DPOT-H**. The table format used in the main experiments of the DPOT paper could serve as a good template.

3.  **Confusing and Unconvincing Scaling Analysis:**
    *   The results in Table 1 are perplexing. It appears to show that by simply increasing the number of parameters, other methods (e.g., U-F2Net, CViT-L) can outperform the proposed C-CSCNN (Large). This raises a critical question: **What is the true significance of equivariance if it can be surpassed by simply scaling up a non-equivariant or less-equivariant model?** The authors do not address this point. To make a convincing case, the authors must demonstrate how their model performs at larger scales (e.g., 100M, 200M, 400M parameters). Can a scaled-up C-CSCNN still maintain a performance advantage over other SOTA models with a similar parameter count? Without this analysis, the benefits of the proposed method remain questionable.

4.  **Uninformative Visualizations:**
    *   Figures 4 and 5, which show signed residuals and rollout errors, are presented without any quantitative context or comparison to the key baselines mentioned above. As they stand, these figures are purely illustrative and provide very little information. Are the residuals for C-CSCNN significantly smaller than for Transolver or CViT-L? How does the rollout error accumulate compared to other methods? These results should be presented in a tabular format with clear metrics (e.g., Mean Squared Error, L2 error at different rollout steps) and compared against the same strong set of baselines used in the main experiments.

**Questions:**

1.  The central claim of the paper is that C-CSCNNs have a "complete kernel basis," yet this is presented as a conjecture (Conjecture 4.1) without a formal proof. While I understand the proof might be non-trivial, could you provide a more rigorous argument beyond the $O(2,0)$ example? For instance, can you show that for any irreducible representations (irreps) of $O(p,q)$, the tensor product of the input and conditioning multivector representations contains all the necessary irreps for a complete kernel basis? Without stronger theoretical backing, the main claim feels unsubstantiated.

2.  Regarding the experimental setup: Why was there such a high degree of inconsistency in the choice of baselines across different experiments? A consistent comparison against a fixed set of state-of-the-art models (e.g., Transolver, CViT-L, DPOT-H) across all tasks would make the results much more convincing. Can you clarify the rationale behind the current experimental design?

3.  Following up on Table 1: The performance of your 55M parameter model is comparable to or worse than several larger models. Could you provide results for a C-CSCNN model scaled to a much larger size (e.g., ~100M or ~200M)? This is crucial to understand if the benefits of equivariance persist at scale or if they are washed out by the sheer capacity of larger, less-structured models.

4.  Could you provide the quantitative results corresponding to Figures 4 and 5 in a table, comparing them directly against the key baselines (Transolver, CSCNN, Swin-Transformer, CViT-L, DPOT-H)? This would allow for a much clearer assessment of the performance improvements.

---

> ### Author Response · Authors · 2025-11-21
> **Response to Reviewer 3dR3**
>
> **Q1:** We kindly note that we already show for the case of O(2) that the geometric product of the input and conditioning multivector contains all the necessary irreps (See Appendix A.4). More specifically, a multivector of Cl(2,0) is a composition of the following irreps: \[0e, 1o, 0o\]. The tensor product of 2 multivectors decomposes into the following unique irreps: (0e, 0o, 1o, 2), where frequency 2 comes from the vector-vector interaction. This results in the analytical basis of Steerable kernels containing frequency 2 components [1], which are absent in the geometric product of a single multivector with itself. In A.4. we demonstrate that for 2 arbitrary multivectors, the frequency 2 components appear in the geometric product, and are therefore recovered.
>
> [1]: Weiler, M., & Cesa, G. (2019). General e (2)-equivariant steerable cnns. Advances in neural information processing systems, 32. https://proceedings.neurips.cc/paper/2019/hash/45d6637b718d0f24a237069fe41b0db4-Abstract.html
>
> **Q2:** We agree that a consistent set of baselines improves the clarity of the comparison. To address this, we have expanded our experimental setup to include the two top-performing SOTA models, Transolver and CViT, on the Navier-Stokes and SWE-1 tasks, using hyperparameters from their original implementations. We did not include DPOT-H, as it was previously shown to underperform relative to CViT (in the CViT paper). We summarised our results in Table 3 of the Appendix, showing that Conditional CSCNNs, when matched by parameter count, outperform all the best performing state-of-the-art baselines in all tasks.
>
> Regarding the original experimental design: our primary goal was to compare with baselines that were previously reported on the specific experiments we used in our evaluation. This strategy ensured we compared against already verified implementations and published results, guaranteeing the fairest assessment of our model.Therefore, less popular experiments such as the 3D Maxwell's equations have relatively few baselines to compare against. Additionally, most current frameworks are not able to handle the relativistic tasks such as our 2D Maxwell's experiment.
>
> **Q3:** We appreciate the reviewer's request to evaluate our model on a much larger scale. Running 100-200M-parameter C-CSCNNs at our spatial resolutions would require substantially more training compute and memory than our current budget allows during the review window. We note however that the contribution of this paper is not to assess the utility of equivariance at large-scale, as this has been investigated in previous works, with evidence that symmetry priors maintain (and enhance) their advantages as models and compute grow \[2\]. For this reason, and because Conditional CSCNNs outperform the strongest baselines under matched parameter counts, we expect their advantage to persist even when scaling to substantially larger model sizes.
>
> \[2\]: Brehmer, J., Behrends, S., De Haan, P., & Cohen, T. (2025). Does equivariance matter at scale? Transactions on Machine Learning Research. <https://openreview.net/forum?id=wilNute8Tn>
>
> **Q4:** We appreciate the reviewer's suggestion. Figures 4 and 5 are intended primarily as qualitative visualizations to illustrate the spatial structure and distribution of prediction errors, which is standard practice in scientific machine learning (e.g. Figure 7 in \[3\], or Figure 3 in \[4\]). Such visual comparisons complement the quantitative analysis by providing intuition about where and how the models succeed or fail.
>
> For the corresponding quantitative performance metrics, we kindly refer the reviewer to Table 3 of the Appendix, which directly compares our proposed model against key baselines.
>
> \[3\]: Wu, H., Luo, H., Wang, H., Wang, J., & Long, M. (2024). Transolver: A fast transformer solver for PDEs on general geometries. In R. Salakhutdinov, Z. Kolter, K. Heller, A. Weller, N. Oliver, J. Scarlett, & F. Berkenkamp (Eds.), Proceedings of the 41st International Conference on Machine Learning (ICML 2024) (Proceedings of Machine Learning Research, Vol. 235, pp. 53681-53705). PMLR. <https://proceedings.mlr.press/v235/wu24r.html>
>
> \[4\]: Wang, S., Seidman, J. H., Sankaran, S., Wang, H., Pappas, G. J., & Perdikaris, P. (2024). Cvit: Continuous vision transformer for operator learning. In International Conference on Learning Representations (ICLR 2025) arXiv preprint: <https://doi.org/10.48550/arXiv.2405.13998>

---

> > ### Comment · Reviewer_3dR3 · 2025-11-24
> >
> > Thank you for your response and clarifications. However, the majority of the critical issues I raised in my review remain unaddressed. Therefore, I will be maintaining my original rating.
> >
> > The core of my concerns could have been directly addressed by conducting the necessary experiments to validate the effectiveness of the proposed algorithm and resolve my doubts. This includes providing a comprehensive comparison against a unified set of baselines, evaluating performance on datasets with other group symmetries, and demonstrating the algorithm's performance at a large number of parameters.
> >
> > The ICLR rebuttal period offers sufficient time to complete at least some of these crucial experiments. However, it appears no significant effort was made in this direction. Consequently, my assessment of this work remains unchanged.
> >
> > Should the authors be willing to supplement their work with the experiments requested in my initial review, I would be happy to engage in further discussion.

---

> > > ### Author Response · Authors · 2025-12-04
> > > **Response to Reviewer 3dR3**
> > >
> > > We thank the reviewer for the additional feedback. In response to the reviewer’s concerns about the effectiveness of our method due to the heterogeneity of the reported results, we conducted additional experiments during the rebuttal period, specifically adding state-of-the-art baselines CViT and Transolver for the SWE-1 and NS tasks. These new results have been incorporated into the revised manuscript and are included in Appendix Table 3, which provides a unified summary of all experiments across all baselines and enables a consistent comparison of our method’s performance.
> > > For clarity, in Table X below we present only the newly added baseline results.
> > >
> > > NS - Test MSE:
> > > | Model            | N = 64   | N = 512  | N = 2048  | N = 4096  |
> > > |------------------|----------|----------|-----------|-----------|
> > > | C-CSCNN (ours)   | 2.38e-03 | 5.80e-04 | 2.70e-04  | 1.80e-04  |
> > > | Transolver       | 2.49e-03 | 7.10e-04 | 4.10e-04  | 3.40e-04   |
> > > | CViT             | 0.0121   | 2.15e-03 | 1.01e-03  | 9.45e-04   |
> > >
> > >
> > > SWE-1 - Test MSE:
> > > | Model            | N = 64   | N = 256  | N = 1024  | N = 2048   |
> > > |------------------|----------|----------|-----------|------------|
> > > | C-CSCNN (ours)   | 1.3886   | 0.3137   | 0.0734    | 0.0519     |
> > > | Transolver       | 1.5127   | 0.4515   | 0.0903    | 0.059404   |
> > > | CViT             | 1.9871   | 0.7129   | 0.1974    | 0.12589    |

---

### Official Review · Reviewer_f8xF · 2025-10-31

**Soundness:** 2
**Presentation:** 1
**Contribution:** 1
**Rating:** 2
**Confidence:** 4

**Summary:**

The paper addresses an incompleteness on the kernel representation defined over the Clifford algebra. The incompleteness is reported at least for the Clifford algebra with characteristics (2, 0), and it has a deteriorating impact on the performance on Clifford neural networks defined with Clifford steerable kernels. The authors introduced (Clifford) group-steerable kernels augmented with auxiliary variables derived from input feature fields to complement the lack of freedom. The proof for the equivariance of the kernels is also given. The experiments are conducted in a range of representative scenarios where the Clifford algebra is known to model well.

Overall, the paper is not ready to be published in the current form, because the significance and indication of the work are unclear. The mathematical notions included in this paper are also very vague and many of the accounts are loose. I expect the paper would need a couple of profound updates and revisions.

**Strengths:**

- Simplicity of the idea. The authors figured out a simple way to make the change to the group steerable kernel. This helps to avoid overly complicating the kernel to be equivariant to the action and also has a knock-on effect for the conciseness of the proof.
-  A conjecture for the completeness is also given in an attempt to promote the research in this field.
- Experiments are conducted in representative scenarios and show an evidence that filling the void of the incompleteness with the conditional kernel could reduce the prediction error.

**Weaknesses:**

**Significance of the work:** While the paper attempts to fill the gap between the theoretical results on the steerable kernel known in the existing literature and its practical implementation, the significance of the theoretical contribution and the empirical results is unclear. I believe this vagueness is stemmed from the lack of satisfactory discussion on applications or scenarios in which this incompleteness causes a crucial problem in downstream tasks. While in Introduction an issue associated with the existing kernel bases is mentioned, they are simply accounted for the "weak expressivity," and it is still unclear when and how this "weak expressivity" is crucial. This also leaves me uncertain that the present work solely focused on $O(2, 0)$ is significant enough, and I also had a difficulty in assessing the significance of the performance improvement in the experiments due to the lack of the discussion. All those aspects leave the overall significance of the work hard to assess.

**Straightforward extension:** The theoretical results do not involve any non-trivial proofs and the majority of them are essentially relying on the same proof as in [1]. Given the unclear motivation mentioned above, the extent of difficulty in giving proof also matters and the theoretical result is relatively weak.

**Mathematical notations:** Mathematical notations are loosely introduced and the paper is very hard to follow. The following is the list of (non-exhaustive) vague points I believe need more detailed accounts:
- Line 113: Mention on the definition of $W$ is missing. I presume, it should be a vector space.
- Equation (2) does not make sense for me. How is the action of $G$ onto the pseudo-Euclidean space formalized here?
- Line 135: Eq. 3.1 (typo)
- Example 3.1 does not follow if we follow the equation (2)
- Line 144: … integral 3.1 … (typo)
- What is det$(g)$? Do the authors assume $G$ is a subgroup in the matrix group?


[1] Zhdanov et.al., Clifford-Steerable Convolutional Neural Networks, ICML 2024.

**Questions:**

Please translate the unclear or uncertain points mentioned in the Weakness column to my questions.

---

> ### Author Response · Authors · 2025-11-21
> **Response to Reviewer f8xF**
>
> We thank the reviewer for the detailed and actionable feedback. We acknowledge the presentation issues and hope to address remaining concerns below.
>
> **W1 (significance of the work):**
>
> We agree that the submission would benefit from further elaborating on the significance of the incompleteness. To address the issue, we now provide a side-by-side visual comparison of the original basis with missing degrees of freedom and the basis generated by conditioned CSCNNs (Ours). One can see in the figure that a part of kernels are simply zero (denoted by gray color), which means that a chunk of the computation is wasted in CSCNNs. On the contrary, in C-CSCNNs, those kernels are non-zero and learn meaningful features, which indeed improves downstream performance, as evident from comparisons on Navier-Stokes, Shallow water equation and Maxwell, where CS-CNNs achieves 2x performance gain. We also would be grateful to the reviewer if they provided ideas to clarify the significance of the submission further if needed.
>
> **Action taken:** We reworked Figure 1 to include side-by-side comparison of CSCNNs and C-CSCNNs (Ours) kernels complemented by empirical gains of our approach on multiple benchmarks.
>
> **W2 (straightforward extension):**
>
> We respectfully disagree with this characterization of our theoretical contributions as trivial. Specifically, we would like to note that the rigorous introduction of conditioned steerable convolutions is novel itself (Section 4.1). Moreover, the derivation of the equivariance constraint of conditioned steerable CNNs (Lemma 4.1) is distinct from any proof in \[1\]. Furthermore, we also derive the constraint on the pooling operator T (Proposition 4.1). While it is not the most involved mathematical proof, it is nonetheless distinct from contributions in \[1\].
>
> **W3 (mathematical notations):**
>
> We are grateful to the reviewer for highlighting the typos in the manuscript. We took care of polishing the manuscript and we hope that it does improve the presentation.
>
> **Action taken:** We fixed the typos in the manuscript.

---

> ### Comment · Reviewer_f8xF · 2025-11-26
>
> Thank you for the reply. I concede that there is a distinction (to some extent) between the proofs of the paper and [1]. I have a follow-up question about the experiments since it is still a bit hard to assess the significance of the contribution.
>
> What class of physical problems would greatly benefit from the work? While the experiments show that the proposed model outperforms the existing models in scenarios where the Clifford-algebra-based representation is well known to fit naturally, I still do not see any other indications beyond this performance gain. I personally think some additional clarification, such as a (sub)set of problems for which the proposed model particularly works well, among or apart from the physical systems the Clifford algebra is known to naturally model, would be very informative to assess the significance of the work; otherwise, the contribution would still look relatively incremental over the well-known range of problems.

---

> > ### Author Response · Authors · 2025-12-04
> > **Response to Reviewer f8xF**
> >
> > We thank the reviewer for the additional feedback. The main advantage of C-CSCNNs is that they respect symmetry transformations of E(p,q) and its subgroups by construction, and can therefore be applied to modelling problems where equivariance is beneficial. Any physical task defined on a (pseudo-)Euclidean space can be straightforwardly embedded into the Clifford algebra representation, which allows for the exploitation of the geometric relationships between scalar, vector, and higher-grade fields.
> >
> > That said, the architecture itself is not restricted to PDE surrogates. Any task with Euclidean or pseudo-Euclidean symmetry and geometric features, such as 3D pose, camera rays, or surface normals in control[1], robotics[2][3] and computer vision[4] can in principle benefit from multivector representations and E(p, q)-equivariance. We provide some example references:
> >
> > [1]: Löw, T., & Calinon, S. (2023). Geometric algebra for optimal control with applications in manipulation tasks. IEEE Transactions on Robotics. arXiv preprint: https://doi.org/10.48550/arXiv.2212.07237
> > [2]: Lin, C. E., Song, J., Zhang, R., Zhu, M., & Ghaffari, M. (2023). SE(3)-Equivariant Point Cloud-Based Place Recognition. In Proceedings of the 6th Conference on Robot Learning (CoRL 2023). Open access: https://proceedings.mlr.press/v205/lin23a.html
> > [3]: Löw, T., Abbet, P., & Calinon, S. (2023). gafro: Geometric Algebra for Robotics. arXiv preprint: https://doi.org/10.48550/arXiv.2310.19090
> > [4]: Chen, H., Liu, S., Chen, W., & Li, H. (2021). Equivariant point network for 3D point cloud analysis. In Proceedings of the IEEE/CVF Conference on Computer Vision and Pattern Recognition (CVPR 2021). arXiv preprint: https://doi.org/10.48550/arXiv.2103.14147

---

### Official Review · Reviewer_BPnS · 2025-11-02

**Soundness:** 1
**Presentation:** 1
**Contribution:** 2
**Rating:** 2
**Confidence:** 3

**Summary:**

This paper proposes Conditional Clifford-Steerable CNNs (C-CSCNNs), an extension to Clifford-Steerable CNNs that aims to address the limited expressivity of the original model. The core idea is to condition the steerable convolutional kernels on auxiliary variables derived from the input feature field, thereby making the kernels input-dependent. The authors provide a theoretical derivation of the equivariance constraint for such conditional kernels and propose an efficient implementation using implicit parameterization and global mean pooling. The method is evaluated on several PDE forecasting tasks, demonstrating improved performance over baseline models.

**Strengths:**

- The paper attempts to ground the proposed method in the established theory of steerable CNNs and Clifford algebra.
- The proposed model demonstrates certain empirical performance.

**Weaknesses:**

- **Lack of Novelty:** The central mechanism of conditioning convolutional kernels on input features is a well-established technique in the broader deep learning literature (e.g., in dynamic filtering and attention). The paper does not demonstrate a fundamental innovation beyond a relatively straightforward application of this idea within the Clifford algebra framework. The conceptual leap from standard conditional convolution to the proposed C-CSCNN is insufficiently articulated.
- **Weak Theoretical Foundation:** The paper's most significant theoretical claim—the completeness of the conditional kernel basis—is presented merely as a conjecture (Conjecture 4.1) without a rigorous proof. For a method that is heavily theorized and whose primary motivation is to solve a theoretical limitation, this is a major shortcoming. The validity of this central claim remains unverified.
- **Inadequate Experimental Analysis and Baselines:**
  - **Lack of Rigorous Ablation:** A critical flaw is the absence of a thorough ablation study. It remains entirely unclear whether the performance gains stem from the sophisticated conditional kernel design or simply from the model having access to a global context vector via mean pooling. Without experiments that ablate the conditioning mechanism itself and compare it to simpler ways of incorporating global information, the claimed 'improved expressivity' remains unsupported and could be an artifact of increased model input information.
  - **Unfair and Unclear Baseline Comparisons:** The paper fails to provide detailed configurations for the baseline models. For instance, it highlights that C-CSCNNs outperform large models like FNO (270M) and UNO (440M), but does not justify **why these baselines require so many parameters**. It is plausible that for the considered tasks, these models were over-parameterized or not optimally configured (e.g., using an excessive number of Fourier modes), which would unfairly inflate the perceived advantage of the proposed method. A fair comparison requires transparent and task-appropriate baseline setups.
- **Major Issues in Presentation and Narrative:**
  - The paper is severely unbalanced. The `Related Work` and `Theoretical Background` sections span over two and a half pages, predominantly reviewing existing knowledge without effectively foregrounding the paper's own contribution. These sections should be significantly condensed and reorganized.
  - Conversely, the `Method` section is underdeveloped. It focuses on mathematical derivations and an idealized case (O(2,0)) but lacks a clear, practical description of the concrete implementation. The narrative fails to guide the reader from the theory to a tangible algorithm, making the work difficult to understand, verify, and replicate. In its current form, the writing quality is unacceptable for publication.

**Questions:**

See Weaknesses

---

> ### Author Response · Authors · 2025-11-21
> **Response to Reviewer BPnS (1)**
>
> We are grateful for the feedback and we hope to address each limitation in the rebuttal.
>
> **W1 (Lack of Novelty):** We respectfully disagree with this characterization. Our contribution is **not** the invention of conditional convolution, which we explicitly acknowledge as established (Section 2, "Implicit (continuous) kernels"). Rather, our contribution is:
>
> - **Identifying a fundamental limitation**: We rigorously demonstrate that CSCNN kernel bases are incomplete (Section 3.3, Example 3.2), missing specific degrees of freedom in the analytical solution.
> - **Deriving equivariance-preserving conditioning**: We derive the mathematical constraints (Lemma 4.1, Lemma 4.2, Proposition 4.1) that conditional kernels must satisfy to maintain G-equivariance while incorporating input-dependent information - this is **non-trivial** and distinct from standard conditional convolution, which does not preserve group equivariance.
> - **Solving the completeness problem**: We demonstrate both theoretically (Example 4.1, Appendix B.5) and empirically (Section 5) that our approach addresses the identified limitation, recovering missing frequency components while maintaining exact equivariance (Table 2: relative error ~10⁻⁷).
>
> To summarize, the novelty lies in **how** we apply conditioning to address a specific theoretical limitation existing in literature, not in claiming to invent the general concept.
>
> **W2 (Weak Theoretical Foundation):** We respectfully but strongly disagree with this assessment, which mischaracterizes both our contributions and the nature of the conjecture.
>
> **Our proven theoretical contributions:**
>
> - **Lemma 4.1**: Derivation of the equivariance constraint for conditional steerable convolution
> - **Lemma 4.2**: Proof that conditional Clifford-steerable kernels satisfy this constraint
> - **Proposition 4.1**: Proof that the conditioning operator preserves equivariance under specified conditions
>
> Those contributions are rigorous and empirically validated -- Table 2 demonstrates equivariance is maintained (relative error ~10⁻⁷). It is those contributions that form the foundation of the work - not the completeness conjecture. Regarding proving the latter, we explicitly acknowledge (page 6, lines 319-323) that this is highly non-trivial. More specifically, it requires:
>
> - Knowledge of all irreducible representations of E(p,q).
> - For each irrep V, we would need to construct an isomorphism between subrepresentations of tensor products of the corresponding Clifford algebra Cl(p,q).
>
> Neither point is trivial. Therefore, proving the conjecture would alone be worthy of a standalone mathematical contribution, from which we abstain in this work, instead focusing on the following validation:
>
> - **Theoretical**: Example 4.1 and Appendix B.5 demonstrate recovery of the missing frequency-2 components for O(2,0), the case where analytical solutions are known.
> - **Empirical**: Section 5 shows consistent improvements over the non-conditioned baseline across four diverse PDE tasks.
>
> To summarize, we believe that characterizing proven equivariance-preserving conditional convolutions as "weak theoretical foundation" because we don't provide a complete representation-theoretic proof is unreasonable and misrepresents the contribution.
>
> **W3 (Lack of Rigorous Ablation):**
>
> We acknowledge the importance of an ablation study and therefore provide it below. Specifically, we ablate the conditioning mechanism on the Shallow-water-1 task. We compare a) non-conditional baseline, b) C-CSCNN with mean pooling, c) C-CSCNN with max pooling, d) C-CSCNN with a fixed multivector drawn from a normal distribution as a condition.
>
> **Additional ablation study (Shallow water 2D, 256 trajectories, MSE):**
>
> - CSCNN (no conditioning): 0.5699
> - C-CSCNN (mean pooling): 0.2982
> - C-CSCNN (max pooling): 0.3097
> - C-CSCNN (random condition): 0.3225
>
> It is evident for the results that mean pooling is indeed the strongest option, followed by max pooling. We note that both mean and max pooling are equivariant, while random condition, on the contrary, is not. The ablation therefore proves the value of equivariance and, in our opinion, provides strong evidence in favour of our framework.
>
> If the reviewer has other suggestions for the ablation, we would be happy to include them in the study. We would also like to clarify regarding increased model input information, which we believe is incorrect. The model receives **identical input** - the same feature field f(x). Pooling is a deterministic, equivariant transformation of existing data, not additional information.

---

> > ### Author Response · Authors · 2025-11-21
> > **Response to Reviewer BPnS (2)**
> >
> > **W4 (Unfair and Unclear Baseline Comparisons):**
> >
> > We kindly note that the baseline results in Table 2 are taken from literature \[Gupta et al., CViT\] and are heavily optimized. In the original work, authors do exhaustive hyperparameter search, and specifically FNO was trained with the assistance of the original author of FNO, Zongyi Li, which is acknowledged in the paper (see Acknowledgements, Gupta et al.). For details, we refer to Table 2 in Gupta et al. which contains exhaustive comparison spanning multiple model sizes and numbers of Fourier modes. Therefore, we believe that the baseline choice is representative and appropriate.
> >
> > **W5 (Major Issues in Presentation and Narrative):**
> >
> > We respectfully disagree with this assessment. We believe the paper's structure appropriately balances theoretical rigor with practical implementation, following standard conventions for ML conference papers. Related works (0.6 page) and Theoretical background (2 pages) are rather typical for the field of geometric deep learning and provide the necessary prerequisites: steerable CNNs (Section 3.1), Clifford algebra and Clifford-Steerable CNNs (Section 3.2), and the specific incompleteness problem we address (Section 3.3). We believe that this background is necessary for readers unfamiliar with Clifford-steerable convolutions, which is a recent framework. If the reviewer has specific suggestions regarding the reformulation, we are willing to consider them, but otherwise we would prefer to keep the structure as it is, given no other reviewer raised it as an issue.
> >
> > Regarding the implementation details, we would again kindly disagree as we dedicate an entire subsection 4.3 to discuss how the method can be done efficiently via template matching. Besides, we discuss experiment-specific implementation details in subsection 5.2. Moreover, we provide implementation details in Appendix B.6, which is a standard way for conference papers: conditional kernel architecture, training setup, dataset specifications. We will also provide code in the camera-ready version.
> >
> > We also find the claim of unacceptable writing quality to be unsupported by specific examples. If the reviewer identifies specific unclear passages, we would gladly revise them. Without concrete feedback, this assessment is not actionable.

---

### Official Review · Reviewer_jhkk · 2025-11-03

**Soundness:** 3
**Presentation:** 2
**Contribution:** 3
**Rating:** 6
**Confidence:** 2

**Summary:**

This paper proposes new theoretical analysis and layer design for Clifford-streerable CNNs. The motivation is that the original CSCNN is not complete in terms of equivariant information. The theoretical analysis eventually leads to a simple design where the layer is additionally conditioned on the average-pooled features. The design conforms with the theoretical insights although the completeness of the simple design is not proved and is formulated as a conjuncture. The model is tested on serval PDE learning tasks including NS, SWE and Maxwell.

**Strengths:**

- The paper includes in-depth theoretical analysis which leads to a simple and pratical design.
- The presented results show improvement against CSCNN baseline.

**Weaknesses:**

- The presentation may not be easily accessible for readers not familiar with CSCNN.
- The performance of the proposed model does not seem to be very competitive on SWE-5.

**Questions:**

- What is the equivariance of the PDE tasks?
- What is the major difference between CSCNN and steerable CNNs? Is the goal of CSCNN also achieving rotation equivariance?
- [1] shows that techniques such as circular padding is important for tasks with periodic boundaries (e.g., SWE). Can C-CSCNN also be adapted for periodic boundary?
- If applicable, you may consider including [1] in the discussion or results since they also use the SWE dataset (also NS).
- In Figure 5, why does the C-CSCNN prediction have a different scale (colorbar) compared to the ground truth?

[1] Sinenet: Learning temporal dynamics in time-dependent partial differential equations. https://arxiv.org/abs/2403.19507

---

> ### Author Response · Authors · 2025-11-21
> **Response to Reviewer jhkk**
>
> We are grateful for the reviewer's positive assessment of our work and hope to address remaining questions with the rebuttal.
>
> Q1: What is the equivariance of the PDE tasks?
>
> - Navier-Stokes and Shallow-water equations are invariant under rotations (SO(N) group), see \[1\], 2.4, p.176.
> - \[2\] demonstrates conformal invariance of Maxwell equations, where Poincaré group R3,1⋊SO(3,1) is a subset.
>
> Q2: What is the major difference between CSCNN and steerable CNNs? Is the goal of CSCNN also achieving rotation equivariance?
>
> - While both approaches are exactly E(n)-equivariant, the main difference is in the type of O(n)-irreps those frameworks handle. Steerable CNNs admit arbitrarily high frequencies, while Clifford-Steerable CNNs only admit irreps with frequencies up to 1 (e.g. in Cl(3,0), vector: 1o, bivector: 1e). The advantage of CSCNNs is that they trivially (just by changing the metric) enable equivariance to the pseudo-Euclidean group E(p,q), which is not so simple to achieve with steerable CNNs (and has not been done, to the best of our knowledge). Therefore, the goal of CSCNN is to achieve not only rotation equivariance, but, if needed, also equivariance to space-time transformations (required in relativistic physics).
>
> Q3: \[1\] shows that techniques such as circular padding is important for tasks with periodic boundaries (e.g., SWE). Can C-CSCNN also be adapted for periodic boundary?
>
> - Yes, indeed! Such adaptation can be achieved straightforwardly by choosing circular padding in the convolutional layer. Furthermore, we tried circular padding on the SWE-5 task (since this is defined on a domain with periodic boundary conditions), but it did not yield substantial gains in performance.
>
> Q4: If applicable, you may consider including \[1\] in the discussion or results since they also use the SWE dataset (also NS).
>
> - We are grateful to the reviewer for pointing out the relevant work and we will include it in the comparison.
>
> Q5: In Figure 5, why does the C-CSCNN prediction have a different scale (colorbar) compared to the ground truth?
>
> - Please note that the colorbar is shared between C-CSCNN prediction and ground truth, so there is no discrepancy.
>
> \[1\] Olver, P.J. (1986). Applications of Lie Groups to Differential Equations.
>
> \[2\] R. M. Wald, General Relativity (Chicago University Press, Chicago, 1984), Appendix D.

---

### Meta-Review · Area_Chair_PEAU · 2026-01-04

**Summary:**

The paper proposes an extension to the Clifford-Steerable CNNs to address a known expressivity issue (namely, the Clifford-Steerable CNNs cannot express certain functions). The paper received one slightly positive score (6) and three negative scores (2). One of the 2s would probably have been raised since the reviewers requested more experiments, which the authors provided.

After reading the paper myself, I believe it has merits: it proposes an interesting idea to fill in a known gap in a popular machine learning architecture. However, I do agree with the reviewers that the paper could be significantly improved. In particular, my main concern is that the paper claims to provide a "complete kernel basis" in its title. However, they do not show that the proposed kernel basis is complete, they only conjecture that, which makes the paper a bit misleading. My suggestion to the authors is to either prove that their architecture is complete (in the space of linear equivariant functions) as they suggested, or give a partial characterization of what functions it can/cannot express. Also, the presentation of the method should be improved for clarity.

**Reviewer Concerns:**

Reviewer jhkk mentioned that the presentation was hard to follow. I agree that some key definitions are missing. In particular it should explicitly say in what space they conjecture their functions to be complete.

Reviewer BPnS mentioned the lack of a theoretical basis for the completeness claim. They also mentioned insufficient experimental evaluation and issues with the presentation.

Reviewer f8xF criticized the significance of the work. The authors gave a good answer to these points.

Reviewer 3dR3 requested further experiments. The authors provided extra experiments in response.

**Reviewer Scores:**

I believe reviewer 3dR3 would have raised their score. Reviewers f8xF and BPnS, I don't know. But on average the score would have remained significantly below the threshold.

---

### Decision · Program_Chairs · 2026-01-26

Reject